# Controlled hydrogenation into defective interlayer bismuth oxychloride via vacancy engineering

Dandan Cui[1,2], Kang Xu[1], Xingan Dong[3], Dongdong Lv[1], Fan Dong [3], Weichang Hao [1✉], Yi Du [1,4✉] & Jun Chen [2✉]

Hydrogenation is an effective approach to improve the performance of photocatalysts within defect engineering methods. The mechanism of hydrogenation and synergetic effects between hydrogen atoms and local electronic structures, however, remain unclear due to the limits of available photocatalytic systems and technical barriers to observation and measurement. Here, we utilize oxygen vacancies as residential sites to host hydrogen atoms in a layered bismuth oxychloride material containing defects. It is confirmed theoretically and experimentally that the hydrogen atoms interact with the vacancies and surrounding atoms, which promotes the separati30on and transfer processes of photo-generated carriers via the resulting band structure. The efficiency of catalytic activity and selectivity of defective bismuth oxychloride regarding nitric oxide oxidation has been improved. This work clearly reveals the role of hydrogen atoms in defective crystalline materials and provides a promising way to design catalytic materials with controllable defect engineering.

[1] School of Physics and BUAA-UOW Joint Research Centre, Beihang University, 100191 Beijing, China. [2] ARC Centre of Excellence for Electromaterials Science (ACES), Intelligent Polymer Research Institute (IPRI) and Australian Institute for Innovative Materials (AIIM), University of Wollongong, Wollongong, NSW 2500, Australia. [3] Research Center for Environmental Science & Technology, Institute of Fundamental and Frontier Sciences, University of Electronic Science and Technology of China, 611731 Chengdu, China. [4] Institute for Superconducting and Electronic Materials (ISEM), Australian Institute for Innovative Materials (AIIM), University of Wollongong, Wollongong, NSW 2500, Australia. ✉email: whao@buaa.edu.cn; ydu@uow.edu.au; junc@uow.edu.au

Semiconducting photocatalysts have attracted tremendous attention due to their capability of utilizing solar light to drive chemical reactions, including water splitting[1], carbon dioxide reduction[2–4] and pollutant degradation[5]. Their performances are determined by electronic structures, surface states and morphologies. The former determines the key charge carrier dynamics in photocatalytic processes including photoexcited charge carrier generation, separation and transport, while the latter dominates the photocatalytic activities. Hydrogenation has been proven to be a promising approach to modify semiconductor-based photocatalysts though modulating physical and chemical characteristics[6]. It can create point defects in photocatalysts that enhance absorption of light by generating impurity bands in the energy gap[7–9]. The defects may also lead to a local electric field that promotes photoexcited charge carrier separation and modified transfer rates, which ultimately enables photocatalysts to achieve excellent catalytic performance[10–12]. In addition, the disordered surface structures induced by hydrogenation can act as active sites for energy conversion reactions[7–9]. As an example, hydrogenation can induce high-density oxygen vacancies in oxide photocatalysts, which facilitates photocatalytic water splitting by broadening the light absorption spectra and modifying the redox ability of photoexcited charge carriers[13–15]. For example, the $TiO_2$ will become yellow or black after hydrogenation and exhibits significantly improved solar photocatalytic performances, and also possess excellent photoelectrochemical–water-splitting performance[13,14]. Recently, theoretical studies revealed that hydrogen can be easily captured by vacancy sites in semiconductors because vacancies always possess high surface energy[16,17]. Protons ($H^+$) can be captured by defects and converted into negatively ionized hydride ions ($H^-$) by migration of electrons, which possibly enhance conductivity and local reactivity of such semiconductors[18,19]. These studies imply that hydrogenation may promote photocatalytic performance of semiconductors not only by creating defects but also through hydrogen doping at these defect sites. Nevertheless, the experimental examination of this hypothesis is limited by technical barriers to visualizing atomic positions and the chemical dynamics of doped hydrogen atoms/ions due to their small atomic radius as well as small atomic mass[20,21]. Whether the doped hydrogen would improve photocatalytic performance, therefore, remains in argument and limits full understanding of hydrogenation mechanisms on photocatalysis.

Bismuth oxychloride BiOCl, a conventional semiconductor photocatalyst, possesses a unique two-dimensional (2D) layered structure, in which the $[Bi_2O_2]^{2+}$ slabs are interleaved with double slabs of halogen ions. This special layered structure can establish an internal electric field to inhibit the recombination of photogenerated charge carriers. This enables BiOCl to efficiently degrade pollutants, decompose water to generate oxygen and reduce the carbon dioxide to carbon monoxide[22,23]. In our previous work, it has been proven that oxygen vacancies can be created easily and are energetically stable, because the material can easily produce a low density of dangling bonds on their surface, which then results in a number of defects. Interestingly, the size of the vacancies are large enough for anchoring and hosting heteroatoms, with hydrogen atoms/ions[24,25]. The hydrogen dopants are likely to be introduced into BiOCl oxygen vacancies by standard hydrogenation approaches. It is, therefore, expected to reveal the role of doping hydrogen atoms by using defective BiOCl as a chemical probe.

In this work, we dope defective BiOCl nanosheets with hydrogen heteroatoms by a low-temperature hydrogeneration method. The hydrogen atoms are successfully anchored at defect sites, which is verified by spectral studies. The NO oxidation activity and selectivity of defective BiOCl are significantly enhanced after hydrogenation. The experimental characterization and theoretical simulation clearly verify that the enhancements of photocatalytic performance are attributable to modulation of energy dispersion by impurity bands, which leads to both improvements in charge separation and photocatalytic activity. This work not only confirms the role of hydrogen heteroatoms in photocatalysts, but also offers a possible way to achieve high-performance photocatalysts via hydrogen doping.

## Results and discussion

**Synthesis and basic characterization of catalyst.** Hydrogenated BiOCl nanoparticles with oxygen vacancies (OV) (H-BiOCl OV) were obtained by annealing BiOCl OV nanoparticles under Ar and $H_2$ atmosphere respectively (experimental details in the supporting information). The morphological feature of as-prepared H-BiOCl OV was characterized by scanning electron microscopy (SEM) and transmission electron microscopy (TEM), as shown in Supplementary Fig. 1, which indicates that the nanoparticles structure shows no apparent change after hydrogenation and no structural disorder or clusters have been detected. The comparison of X-ray diffraction (XRD) patterns (Supplementary Fig. 2) demonstrate that the H-BiOCl OV was of high purity with orientation along the [001] direction (JCPDS card no. 73-2049), indicating that the nanoparticle crystal structure was not changed after hydrogenation. In the high-angle annular dark-field scanning transmission electron microscopy (HAADF-STEM) image (Fig. 1b), we can clearly see the high crystallinity of the nanoparticles and the interlayer distance is about 0.76 nm ($d_{001}$) in H-BiOCl OV, which is very similar to BiOCl OV (0.78 nm). These results suggest the hydrogen atoms have diffused into the oxygen vacancies and settled down with the vacancies; meanwhile, structural distortion or further disorder is not expected, as illustrated in the schematic diagram in Fig. 1a.

Furthermore, the effect of hydrogen atoms on the chemical bonding, surface compositions and light absorption is examined. As shown in Supplementary Fig. 3a, b, compared to the BiOCl OV, the UV−Vis diffuse reflectance spectrum of the H-BiOCl OV shows stronger visible light absorption and a small redshift in extrapolated absorption edge. This suggests that the oxygen vacancies that induce the visible light absorption of BiOCl OV have been effectively mediated by hydrogen atoms[23,25,26]. This is also confirmed by the electron paramagnetic resonance (EPR) measurement, which can probe the electrons trapped in the oxygen vacancies[23,27]. As shown in Fig. 1c, the BiOCl OV exhibited a strong EPR signal at $g = 2.004$ than H-BiOCl OV, indicating that H-BiOCl OV traps a lower number of electrons in oxygen vacancies[2]. At the same time, as illustrated by the $O_{1S}$ X-ray photoelectron spectroscopy (XPS) spectra in Fig. 1d, the peak at 531.13 eV could be ascribed to surface hydroxyls oxygen atoms, which is in the shadow of oxygen vacancies.

The peak of H-BiOCl OV exhibits a 0.15 eV blue-shift, which means the electrons around the oxygen atoms have been reduced due to the inductive effect of hydrogen atoms. According to the peak area at 531.13 eV of $O_{1s}$, we can conclude that the hydrogen atoms could result in oxygen vacancies being reduced and the number of surface hydroxyl oxygens decreased[2,24]. Meanwhile, the distribution of electrons results in the $Bi_{4f}$ XPS peaks of H-BiOCl OV being blue shifted by 0.23 eV compared with BiOCl OV (Fig. 1e). This is further confirmed by extended X-ray absorption fine structure (EXAFS) spectroscopy. The EXAFS spectra of Bi $L_3$-edge were further performed to investigate the valence and coordination states of Bi in samples[28]. Supplementary Fig. 4 shows the absorption edge of H-BiOCl OV moves to lower energy than BiOCl OV, indicating that the lower states of Bi appear. In the Fourier transform spectra of the Bi $L_3$-edge EXAFS oscillations (Fig. 2b), we can see there is no shift of Bi-O at the sharp peaks around 1.6 Å, but the peak of Bi-Bi

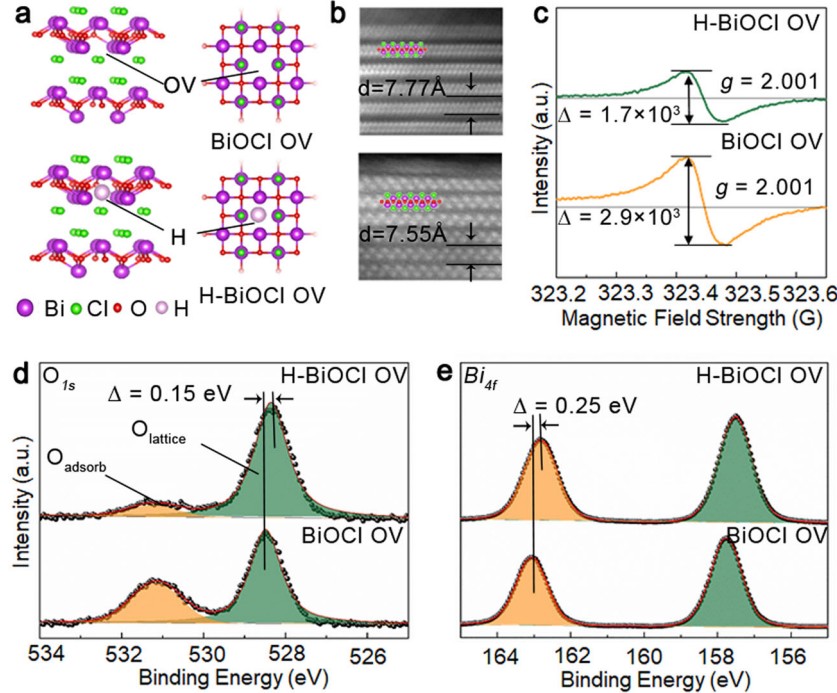

**Fig. 1 Crystal structure and defects characterization of catalyst. a** Crystal structures of BiOCl OV and H-BiOCl OV. **b** High-resolution dark-field STEM image of H-BiOCl OV and BiOCl OV. **c** EPR spectra of all the samples. **d** X-ray photoelectron spectroscopy (XPS) spectra of $O_{1s}$ (**d**) and $Bi_{4f}$ (**e**).

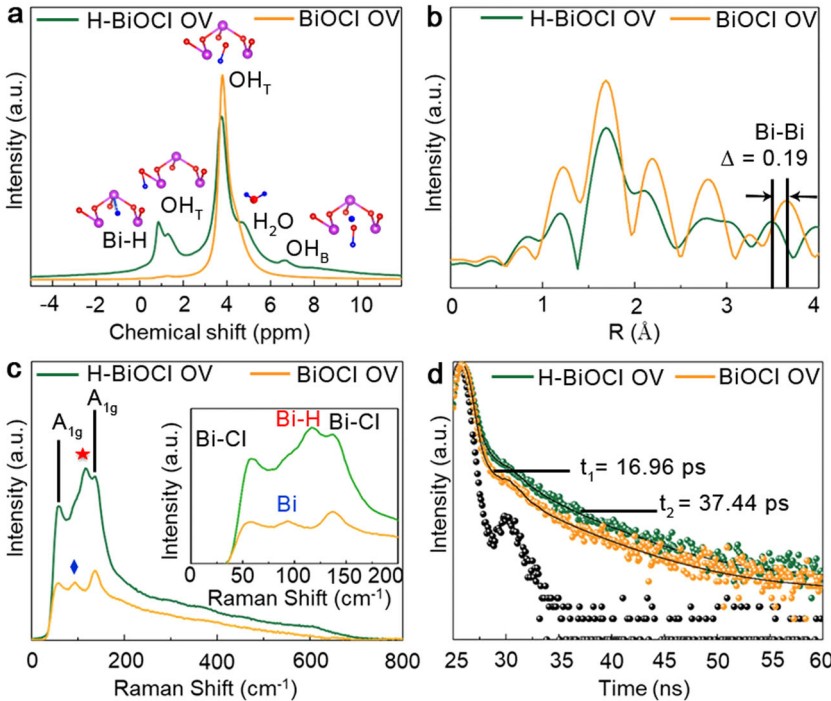

**Fig. 2 Structural characterization of the synthesized powders. a** Comparison of 1H NMR spectra of H-BiOCl OV and BiOCl OV. **b** Experimental Fourier transform of the Bi $L_3$-edge EXAFS. **c** Raman spectra of all the samples. **d** Time-resolved spectrum of prepared BiOCl OV, H-BiOCl OV under 338 nm excitation using 1 ns diode laser and emission wavelength fixed at 430 nm.

at 3.8 Å in H- BiOCl OV is shorter than in BiOCl OV. It indicates that there are a number of localized electrons around Bi atoms and the orbitals of hydrogen atoms simultaneously form a hydrazide with bismuth atoms[25], which validates the localized charge density around vacancy sites being decreased due to the presence of the hydrogen atoms.

**Structural characterization of hydrogen-modified defective BiOCl.** The location of hydrogen atoms in H-BiOCl OV was studied by solid-state nuclear magnetic resonance spectroscopy (ssNMR) measurements, which can confirm the location of hydrogen atoms in our sample. Figure 2a shows the 1H solid-state NMR spectra of BiOCl OV and H-BiOCl OV. The main peak at

the chemical shift of 3.78 ppm is about the main type of bridging proton on the surface, which can be assigned to the terminal hydroxyl groups ($OH_T$) and water molecules associated with the surface sites[29,30]. The slightly larger peak at 4.52 ppm and 4.76 ppm is surface-adsorbed water[31]. It is worth noting that an additional small peak centered at the chemical shift of 6.68 ppm is observed in H-BiOCl OV. It can be assigned to the 1H signal of bridging hydroxyl groups ($OH_B$) in hydrogen-bonding interaction with surface-adsorbed $H_2O$, which is the crucial group to affect the photocatalytic reaction process[29]. The minor peak around 0.8 ppm corresponds to hydrogen atoms with a weakened shielding effect from their surrounding environment, and these can be attributed to internal bridging hydrogen atoms (Bi-H-Bi) where the hydrogen atoms occupy the oxygen vacancies[32]. At the same time, H-BiOCl OV has a minor peak at 1.31 ppm, which can be associated with a typical chemical shift region for the terminal hydrogen atoms ($OH_T$)[32]. These two peaks are probably related, with the hydrogen atoms located in the more complicated environments, suggesting that the hydrogen atoms' mobility in H-BiOCl OV is higher than that of BiOCl OV. It agrees with the Fourier transform infrared (FT-IR) spectroscopy.

As shown in Supplementary Fig. 5, both materials show similar absorption features from 500 to 4000 cm$^{-1}$, with an absorption peak at 1604 cm$^{-1}$ in the FTIR spectrum ascribed to bending vibrations of free water molecules, signifying isolated hydroxyl groups with O-H deformation vibrations. The peaks at around 3470 cm$^{-1}$ are associated with the O-H stretch of intermolecular hydrogen bonds due to O–H stretching and wagging modes[6,33]. The intensity of absorption band around 3400 cm$^{-1}$ of H-BiOCl OV is weaker and wider than BiOCl OV, which indicates that the hydrogen incorporated into the BiOCl OV possibly does not passivate a significant number of OH dangling bonds as this would otherwise increase the absorption[30]. This suggests that the OH groups experience a more varied environment on the H-BiOCl OV surface rather than on the surface of the BiOCl OV.

In order to get further understanding of the hydrogen atoms in the sample, we obtained Raman spectra of BiOCl OV and H-BiOCl OV (Fig. 2c). The peaks at 60.9 and 136.25 cm$^{-1}$ can be ascribed to the $A_{1g}$ internal Bi−Cl stretching and the $E_g$ external stretching modes of Bi−Cl[25]. It has been widely recognized that the lattice periodicity and symmetry breakage can induce new vibration modes. As a result of the existence of oxygen vacancies in the materials, there is a new peak at 99.2 cm$^{-1}$ that appears in the spectra of BiOCl OV and H-BiOCl OV, which corresponds to $A_{1g}$ phonon modes of rhombohedral Bi[34]. Meanwhile, the new peak at 116.3 cm$^{-1}$ appears in the spectra of H-BiOCl OV. This cannot be ascribed to any ordinary Raman bands of BiOCl, and may originate from surface or intersurface vibration modes induced by the hydrogen atoms[35]. Compared with the NMR result we think this new peak may relate to the interaction between hydrogen atoms and bismuth atoms.

## Density functional theory calculations and experiment characterization.
To uncover the effect of hydrogen atom in mediating the vacancies and the electronic structures, density functional theory (DFT) calculations were carried out. As shown in Fig. 3a, b, the valence band maximum (VBM) of H-BiOCl OV is upshifted to the Fermi level ($E_f = 0$ eV) than BiOCl OV, with both samples showing indirect band gaps. The electronic density of states (DOS) clearly suggests that the new electronic states between the gap of the BiOCl OV are mainly created by

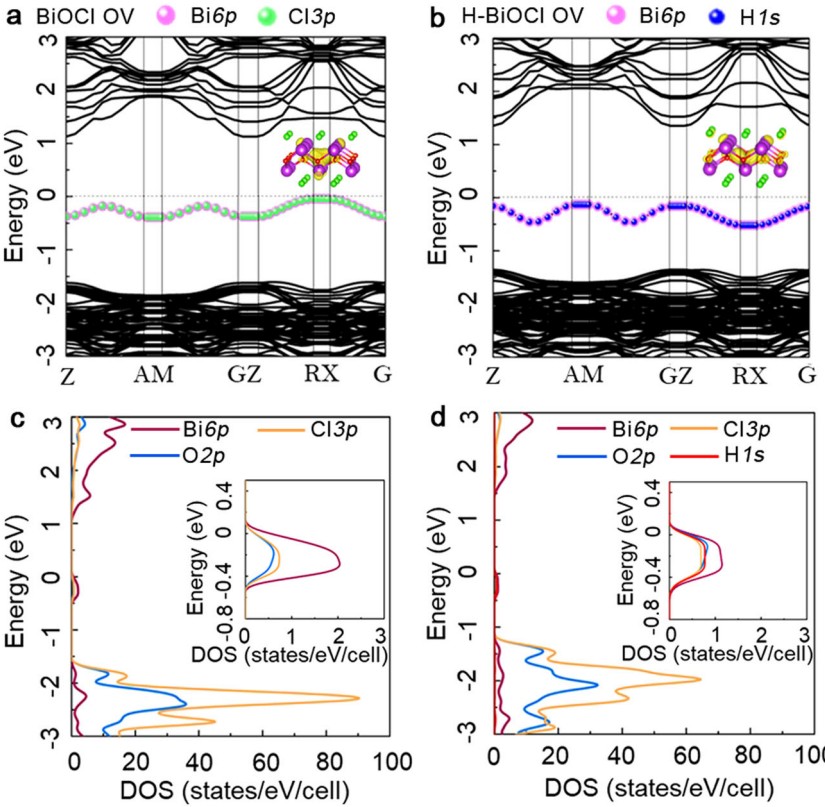

**Fig. 3 Density functional theory calculations of electronic structure for BiOCl OV and H-BiOCl OV.** Band structure of BiOCl OV (**a**) and H-BiOCl OV (**b**), which gives the corresponding charge densities of defect energy levels as inserts and the pink, green, blue colors in bands represent the contributions of Bi6p, Cl3p, H1s orbitals. Density of states (DOS) of BiOCl OV (**c**) and H-BiOCl OV (**d**), the enlarged inserts are the PDOS of defect states. Red, orange, blue, pink lines are Bi6p, Cl3p, O2p and H1s orbitals.

hybridization of occupied Bi-$6p$, Cl-$3p$, and O-$2p$ orbitals (Fig. 3c, d). But in the H-BiOCl OV, the defect states are mainly composed of the hybridization of Bi-$6p$ and H-$1s$. Meanwhile, we have obtained the reaction energy path of the hydrogen atom diffusion into the BiOCl OV. The detailed atomic configurations of the initial state, intermediate states, and the final state are also plotted in Supplementary Fig. 6. We found that the hydrogen atom only needs to overcome an activation energy barrier around 0.15 eV from the free standing BiOCl OV surface to migrate to a position which is near to the oxygen defect location. The final energy state is 1.75 eV lower than the initial state. The charge density of the defect state analysis indicates that electrons move from OVs to the neighboring Bi and O atoms in the BiOCl OV (Fig. 3a, inset image). When an H atom stays on the site of OV, the neighboring Bi and O atoms give electrons to the H atom, making the hydrogen atom negatively charged (Fig. 3b, inset image). More importantly, as shown in Supplementary Fig. 7, the charge density difference map shows the displacement of electronic charge induced by the interaction of the H atom with its Bi neighbor. It indicates that charge accumulates on the H atom and is depleted on the neighboring Bi atoms, and the overlapping of charge density of hydrogen and the closest O atoms can be observed. This implies that there is a bonding cloud between H atom and Bi atoms. These results agree well with the experimental results presented above.

Additionally, from the above theoretical simulations, we can see that the hydrogen atoms can affect the material band structure. Through a plot of the transformed Kubelka–Munk function versus light energy (Supplementary Fig. 3b), the bandgap of H-BiOCl OV is 2.88 eV, which is narrower than BiOCl OV (2.99 eV). The XPS valance band spectra (Supplementary Fig. 8a) and Mott–Schottky plot (Supplementary Fig. 8b) were employed to directly pinpoint the valence band position of the semiconductor. In detail, the VBM and the flat band potential of H-BiOCl OV is upshifted from BiOCl OV. As displayed in Supplementary Fig. 9a, the optical fluorescence spectra of BiOCl OV exhibits two main peaks centered at 434 and 514 nm. Those two main peaks come from the intrinsic band-edge and defect state emissions. Comparing with the BiOCl OV, the second peak's position of H-BiOCl OV is red-shifted and the ratio of two main peak areas becomes bigger. This means the intermediate states in the bandgap of H-BiOCl OV are upshifted and broader than BiOCl OV. The schematic illustration of the band structure of BiOCl OV and H-BiOCl OV has been shown in Supplementary Fig. 9b, which indicates that the VBM of H-BiOCl OV is much more positive than BiOCl OV, which will make it energetically feasible for the trapped electrons to generate $O_2 \bullet^-$. The resulting defect trap states can allow the electrons to be easily photoexcited to the conduction band (CB) from valence band (VB) and transferred to the surface. Hence, the electrons will be abler to activate $O_2$ to $O_2 \bullet^-$ and OH to OH$\bullet$. Furthermore, the electron spin resonance (ESR) spectra (Supplementary Fig. 10) suggest that H-BiOCl OV will possess a better ability than BiOCl OV to activate $O_2$ to $O_2 \bullet^-$ after visible light irradiation, implying that the induced hydrogen atoms can enhance charge carrier yields.

To further understand the effect of hydrogen atom-occupied vacancies on the charge separation inside the semiconductors, we performed photoluminescence measurements. The time-resolved fluorescence emission decay spectra are shown in Fig. 2d, indicating that the H-BiOCl OV exhibited the longer lifetime of ($\tau_1 = 1656.56$ ps, 1.95%, $\tau_2 = 277.00$ ps, 97.40%, $\tau_3 = 6019.14$ ps, 6.3%, average lifetime ~37.44 ps) than BiOCl OV ($\tau_1 = 2016.77$ ps, 7.57%, $\tau_2 = 248.40$ ps, 98.99%, $\tau_3 = 6803.89$ ps, 2.47%, average lifetime ~16.81 ps). In addition, the sample exhibits a transient photoresponse to visible light, as shown in Supplementary Fig. 11a, which clearly indicates that the H-BiOCl OV can generate more photoinduced charge carried by absorbing visible light than BiOCl OV. Moreover, it is interesting to find that the arc radius on the EIS Nyquist plot (Supplementary Fig. 11b) of H-BiOCl OV is smaller than the arc radius of BiOCl OV under visible light irradiation, which suggests that the H-BiOCl OV has higher efficiency in separating and transferring photogenerated electron−hole pairs among the interface. Accordingly, using the hydrogen atom-mediated vacancies can reduce the photogenerated recombination rate and increase the ability of redox reactions.

**Photocatalysts performance characterization**. To verify the above supposition and shed light on the effect of hydrogen atom introduced into the oxygen vacancies on the photocatalytic process, photocatalytic nitric oxide (NO) oxidation of both BiOCl OV and H-BiOCl OV was measured under visible light ($\lambda >$ 420 nm) from a 100 mW commercial tungsten halogen lamp. As depicted in Fig. 4a, the H-BiOCl OV reaches a high NO removal ratio of 45% after irradiation for 30 min, which is about 13% more efficient than BiOCl OV. In addition, the corresponding conversion rates of H-BiOCl OV for generating $NO_X^-$ is much greater than BiOCl OV (Supplementary Fig. 12) in the whole catalytic process. As shown in Supplementary Fig. 13, H-BiOCl OV can be efficiently reused for NO removal with good recyclability and stability, which did not show any loss of photocatalytic activity, and the XRD patterns of reused H-BiOCl OV (Supplementary Fig. 14) do not show any obvious variation after cycling photooxidation test. These results clearly show that H-BiOCl OV exhibit good photocatalytic recyclability and stability under visible light.

To further uncover the underlying reasons for the enhancement of NO removal efficiency by H-BiOCl OV over BiOCl OV, in situ FT-IR spectroscopy measurements were performed on two samples of each material. The in situ FTIR spectra (Supplementary Fig. 15) show that the absorption peaks of NO (963, 1612 cm$^{-1}$)[36], $NO_2$ (1443 cm$^{-1}$)[37] and $NO_2^-$ (1174 cm$^{-1}$)[38] appear on the surface of both H-BiOCl OV and BiOCl OV after they adsorb NO in the dark for 20 min. The intensity of all absorption bands on the surface of H-BiOCl OV is stronger than BiOCl OV (Fig. 4c). From the in situ FTIR spectra of photocatalytic reaction processes on the surface of H-BiOCl OV (Fig. 4b) and BiOCl OV (Supplementary Fig. 16a) under visible light irradiation, we see that new absorption bands of $NO_2$ (at 1725 cm$^{-1}$)[39] and nitrates (at 809, 1275 and 1494 cm$^{-1}$)[40] appear and increase during the whole light irradiation process (Fig. 4d), indicating that an amount of nitrite and nitrate ions is produced on the catalyst surface.

It is worth noting that the strong characteristic peak intensity of $\bullet O_2^-$ radicals at 1003 cm$^{-1}$ is typically indicated on the surface of the samples[41]. The absorption intensity of this intermediate product has been normalized (Supplementary Fig. 16b). The change trends of this reactive oxygen species on the surface of BiOCl OV and H-BiOCl OV are similar. The final absorption intensity of $\bullet O_2^-$ by H-BiOCl OV is stronger than that of BiOCl OV; therefore, the $O_2$ molecules adsorbed on the surface of H-BiOCl OV are more easily activated and can receive more electrons to convert to $\bullet O_2^-$ radicals that adsorb on the surface of BiOCl OV. This reactive oxygen species participates in photocatalytic redox reaction, which ultimately enhances the efficiency of the photocatalytic reaction.

At the same time, the absorption bands located at 2162 cm$^{-1}$ (NO$^+$)[42] appear on the surface of both these sample types. The normalized absorption of this product reveals that the relative content of NO$^+$ on the surface of H-BiOCl OV is larger than on BiOCl OV (Supplementary Fig. 16c). Moreover, we found the 1673 cm$^{-1}$ absorption band (NO$_2^+$)[43] only appears on the

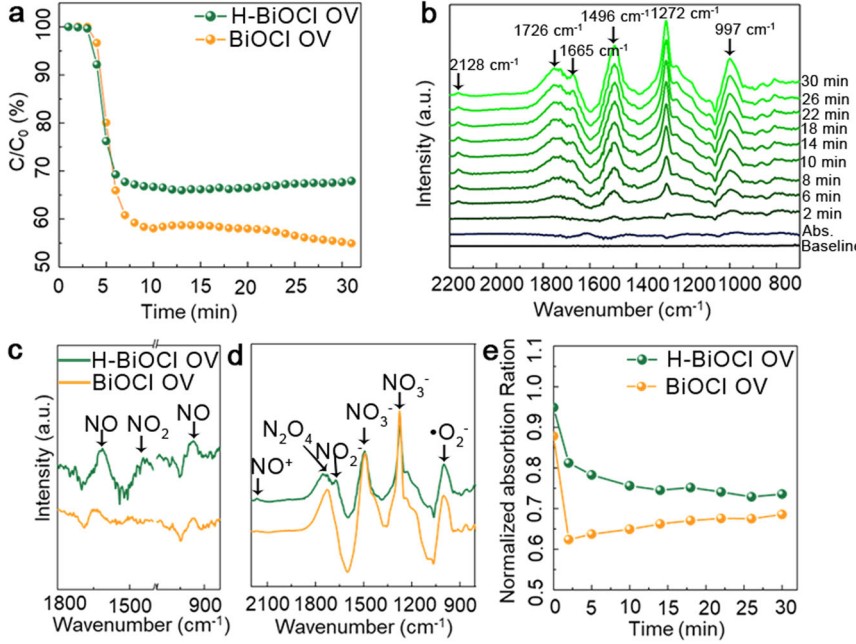

**Fig. 4 Photocatalytic activities of the sample. a** Photocatalytic removal of NO in the presence of BiOCl OV and H-BiOCl OV under illumination by visible light ($\lambda > 420$ nm). **b** In situ FTIR spectra of the photocatalytic NO oxidation process over H-BiOCl-OV under simulated solar-light irradiation. **c** In-situ FTIR spectra of the NO adsorption process (**d**) and degradation process. **e** Corresponding normalized absorbance radio between $\nu$ ($\bullet NO_3^-$) 1003 cm$^{-1}$ and $\nu$ ($NO_2$) 1725 cm$^{-1}$.

surface of H-BiOCl OV (Fig. 4b). Those two products are derived from oxidation of NO and $NO_2$ by $\bullet OH$ and $\bullet O_2^-$, then final oxidation into the nitrites or nitrates. In addition, $NO_2$ is a toxic air pollutant, so the conversion of NO by oxidation into $NO_2$ should be inhibited. Comparing the normalized absorption intensity of the $NO_3^-$ (1274 cm$^{-1}$) and $NO_2$ (1725 cm$^{-1}$), the proportions on the trend chart of Fig. 4e indicate that the H-BiOCl OV produces more $NO_3^-$ rather than $NO_2$. Overall, these results demonstrate that utilizing hydrogen atoms to modify oxygen vacancies can directly and immediately enhance the reactive efficiency and selectivity of BiOCl OV.

The possible reaction mechanism of NO photocatalytic oxidation by H-BiOCl OV we propose is given in the Supplementary Discussion.

In summary, we successfully introduced hydrogen atoms into a defective BiOCl crystal structure by utilizing the oxygen vacancies. By integrating the results of the experiments and theoretical calculations, we have demonstrated the role of hydrogen atoms in the BiOCl lattice with O vacancy defects. The hydrogen atoms show a preference to occupy the oxygen vacancies' states and hybridize with the nearby atoms. This results in generation of new trap states among the impurity states and upshifts the position of the valence band. This effect improves the redox reactivity and selectivity efficiency of BiOCl OV for solar-driven NO oxidation by improving the separation and transfer efficiency of photogenerated carriers. This study provides distinct insights exploring the role of hydrogen atoms in oxide materials having defects. Importantly, this work not only provides a feasible plan for designing high-efficiency photocatalysts through introduction of simple elements, but also sheds light on the crucial role of elemental doping in photocatalysts at the atomic level.

## Methods

**Catalyst characterization**. Powder XRD patterns were collected via an X-ray diffractometer (GBC MMA diffractometer) with Cu Ka radiation and a working voltage of 40 kV. The morphologies of the as-prepared samples were characterized by SEM (JEOL JSM-7500FA). The details of the crystal structure were further examined by STEM (JEOL JEM-ARM 200 F, operating at 200 kV). UV−Vis diffuse reactance spectra were collected on UV–Vis–NIR spectrophotometer (UV-3600, Shimadzu) using 100% BaSO$_4$ as the reference sample. X-ray spectroscopy (XPS) and X-ray absorption spectra (XAS) were conducted at the Photoelectron Spectroscopy Station (Beamline 4W9B) of the Beijing Synchrotron Radiation Facility of the Institute of High Energy Physics, Chinese Academy of Sciences. EPR and ESR spectroscopy were conducted on a JES FA-200 spectrometer. FT-IR spectrometry was obtained from a Shimadzu FTIR Prestige-21. The Raman spectra were obtained by the Nanofinder system. The 1H NMR spectra were acquired on a Bruker Advance III 500WB spectrometer. Photoluminescence data were collected using a Jobin Hvon Flurolog 3 from Horiba, with a Xenon arc lamp as a light source and either PMT or InGaAs detector for visible and NIR collection, respectively. The time-resolved phosphorescence spectra were collected using a 334 nm Nano LED source as the examination light.

**Photoelectrochemical characterization of catalyst**. A three-electrode cell was used to carry out the electrochemical measurements on an electrochemical workstation (CHI-660D, China). The working electrode was a catalyst-coated FTO electrode. An Ag/AgCl electrode was used as the reference electrode. A Pt wire was used as the counter electrode and saturated 0.1 M Na$_2$SO$_4$ solution was used as the electrolyte. A 300 W Xe lamp with a cut-off filter ($\lambda > 420$ nm) was used as the light source. The Mott−Schottky measurements were monitored at a fixed frequency of 100 Hz with 10 mV amplitude at various potentials.

**Photocatalytic reactions and in situ FTIR investigation**. The photocatalytic activity was evaluated based on the removal efficiency of NO at ppb levels in a continuous flow reactor with 0.2 g prepared sample. The concentration of NO was continuously detected by an NOx analyzer (42c-TL, Thermo Environmental Instruments Inc.). A 150 W commercial Xenon lamp with a 420 nm cut-off filter that was vertically placed above the reactor glowed when the adsorption–desorption equilibrium was achieved. The quantum efficiencies of NO oxidation at a variety of wavelengths were measured by inserting monochromatic filters in front of the reactor. In situ FTIR measurements were conducted using a TENSOR II FT-IR spectrometer (Bruker) equipped with an in situ diffuse reflectance cell (Harrick) and a high-temperature reaction chamber.

The reaction chamber was equipped with three gas ports and two coolant ports. High-purity He, high-purity O$_2$, and a mixture of 100 ppm of NO in He could be fed into the reaction system, and a three-way ball valve was used to switch between the target gas (NO) and the purge gas (He). The total gas flow rate was 100 mL min$^{-1}$, and the concentration of NO was adjusted to 50 ppm by dilution with O$_2$. The chamber was enclosed with a dome having three windows, two for IR light entrance and detection, and one for illuminating the photocatalyst. The observation window was made of UV-absorbing quartz and the other two windows were made of ZnSe. A Xenon lamp (MVL-210, Optpe, Japan) was used as the irradiation light source.

Before the measurements, the prepared products were placed in a vacuum tube and pretreated for 1 h at 200 °C.

**DFT calculations**. A DFT computational study of the electronic structures was carried out using a Simulation Package (VASP). The VASP package implemented DFT in the Kohn–Sham formulation using a plane wave basis and the projector-augmented wave formalism (PAW). Bi-6$p$, Cl-3$p$, and O-2$p$ electrons were treated as the valence electrons in the PAW potentials. A $2 \times 2 \times 2$ supercell of monoclinic phase BiOCl was considered in the calculations. All atoms were fully relaxed for structural relaxation until the atomic forces were smaller than 0.01 eV Å$^{-1}$ on each atomic site. The energy cut-off of the plane wave basis set was 500 eV. $2 \times 2 \times 2$ $k$-points were employed in the calculations. For the structure containing an O defect, an oxygen atom was removed from the BiOCl supercell. For the H-addition simulations a hydrogen atom was added into the BiOCl OV supercell.

**Syntheses and materials**. Details regarding the chemicals and gases used, as well as catalyst syntheses are included in the Supplementary Methods.

## Data availability
The authors declare that the data supporting the findings of this study are available within the article and Supplementary Information, or from the corresponding author upon reasonable request.

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

## Acknowledgements
This work was financially supported by the National Natural Science Foundation of China (Grant Nos. 51672018, 11874003), and the Academic Excellence Foundation of BUAA for Ph.D. Students, Fundamental Research Fund for Centre University and Australian Research Council (ARC) Centre of Excellence Scheme (CE140100012&DP170102267 &DP170101467). The authors would like to thank the scholarship support from China Scholarship Council (CSC).

## Author contributions
D.C., W.H., Y.D. and J.C. conceived and designed the research. D.C. performed the synthesis and carried out all the catalytic tests. K.X. contributed to DFT calculations. D.L. performed the NMR test. X.D. and F.D. performed the in situ FTIR test. All authors participated in drafting the paper and gave approval to the final version of the manuscript.

## Competing interests

The authors declare no competing interests.
