## [Peer Review File · Communications Chemistry]

Reviewers' comments:

Reviewer #1 (Remarks to the Author):

In the manuscript, the mechanism of hydrogenation and synergetic effect between hydrogen atoms and oxygen vacancies of BiOCl photocatalysts have been studied by various experimental and theoretical methods. The authors utilized oxygen vacancies as residential sites to host hydrogen atoms in BiOCl with oxygen vacancies in the hydrogenation, and found that the hydrogen atoms interact with the vacancies and surrounding atoms, which promotes the separation and transfer processes of photogenerated carriers via the resultant band structure. This is a novel and effective approach to improve the activity and selectivity of the BiOCl photocatalysts. I recommend the publication of the manuscript after taking the following comments into account.

According to the NMR and Raman data given by the authors, I also agree that the Bi-H site is present in H-BiOCl OV, but it does not guarantee that H has entered the original OV. There are other possibilities. For example, the H replace the hydroxyl on the surface of the catalyst to form the Bi-H site for hydrogenation, and/or the surface hydroxyl is dehydrated to form new defects (such as bridging oxygen) in the hydrogenation. All these may also be the reasons for the decrease in terminal hydroxyl groups and the appearance of bridged hydroxyl groups (see XPS and NMR spectra). I suggest that the authors should compare the structural changes before and after hydrogenating BiOCl without or with a few oxygen vacancies. If there is almost no formation of OV and Bi-H sites, it can be proved that H does enter in the original OV of the BiOCl OV photocatalysts. Additionally, the legend of Figure 2 and 4 should be under the Figure.

Reviewer #2 (Remarks to the Author):

As we all know, the exploration of photocatalytic reaction mechanism associated with defect engineering is a basic issue for designing and developing of photocatalytic semiconductor materials. This paper utilizes a variety of methods to systematically study the mechanism of hydrogenation and synergetic effects among hydrogen atoms and local electronic structures in the defective interlayer BiOCl. This study is profound and systemic, and I will recommend the manuscript for publication with minor revision.

- (1) The excellent recyclability and stability of a catalyst should be very important. Please the authors perform relevant tests to verify their as-prepared samples.
- (2) Please confirm whether the sample structure changes after a long-time test.
- (3) About the EXAFS. The absorption edge of Bi4f maybe also change when the electrons around Bi4f change. Can you supply the raw data before the Fourier transform?
- (4) The reaction mechanism is important for catalyst. Can you provide the possible reaction mechanism of NO photocatalytic oxidation for H-BiOCl OV.
- (5) The authors should compare and review with the relevant data of previously reported literatures of BiOCl OV (Comp. Mater. Sci., 2012, 61: 180-184; ACS Catalysis, 6(12): 8276-8285; J. Mater. Chem. A, 2017, 5(47): 24995-25004; Appl. Catal. B-Environ.: 2018, 228: 87-96, etc.).

Reviewer 1

In the manuscript, the mechanism of hydrogenation and synergetic effect between hydrogen atoms and oxygen vacancies of BiOCl photocatalysts have been studied by various experimental and theoretical methods. The authors utilized oxygen vacancies as residential sites to host hydrogen atoms in BiOCl with oxygen vacancies in the hydrogenation, and found that the hydrogen atoms interact with the vacancies and surrounding atoms, which promotes the separation and transfer processes of photogenerated carriers via the resultant band structure. This is a novel and effective approach to improve the activity and selectivity of the BiOCl photocatalysts. I recommend the publication of the manuscript after taking the following comments into account.

1. Comment

According to the NMR and Raman data given by the authors, I also agree that the Bi-H site is present in H-BiOCl OV, but it does not guarantee that H has entered the original OV. There are other possibilities. For example, the H replace the hydroxyl on the surface of the catalyst to form the Bi-H site for hydrogenation, and/or the surface hydroxyl is dehydrated to form new defects (such as bridging oxygen) in the hydrogenation. All these may also be the reasons for the decrease in terminal hydroxyl groups and the appearance of bridged hydroxyl groups (see XPS and NMR spectra). I suggest that the authors should compare the structural changes before and after hydrogenating BiOCl without or with a few oxygen vacancies. If there is almost no formation of OV and Bi-H sites, it can be proved that H does enter in the original OV of the BiOCl OV photocatalysts. Additionally, the legend of Figure 2 and 4 should be under the Figure.

Response

We thank for this suggestion. Obviously, the H can replace the hydroxyl on the surface or dehydrated to form new defects (such as bridging oxygen) in the hydrogenation¹. Regarding the hydrogen atom states, we cannot guarantee all the H atoms have entered the original OV. But through theoretical research, we can be sure that H can enter the original OV of the BiOCl OV photocatalysts.

The properties of atomic hydrogen in semiconductors have been widely investigated. In the high electron-density regions, the H tends to lose its electron by behaving as a donor and forming H^+ ions. In regions of low density, instead, H can gain one electron by assuming the stable configuration of helium and forming H^- ions, thus behaving as an acceptor. Therefore, H can form stable H–VO complexes where it takes the place of the missing O atom by forming a bond with a prevailing ionic character^{2,3}.

For example, in the ZnO, the theory and experiment show that the H forming a complex with an oxygen vacancy, where it takes the place of the missing O atom and binds covalently to the surrounding Zn atoms^{4,5}.

Through the theoretical calculations, we obtain the reaction energy path of the hydrogen atom diffuse into the BiOCl OV. As shown in Supplementary Figure 6. We found that the hydrogen atom only needs to overcome an activation energy barrier about 0.15 eV from the free-standing BiOCl OV surface to migrate to a stable position, in which the hydrogen atom can stably tap into the oxygen defect location. At the same time, when the H atom stays on the site of OV, the charge density difference map (Supplementary Figure 7) shows the displacement of electronic charge induced by the interaction of the H atom with its Bi neighborhood. This implies that there is a bonding cloud between the H atom and Bi atoms. These results agree well with the NMR and Raman experimental results.

Response letter

Supplementary Figure 6 Reaction energy paths for H atom dissociation, upper panel, and H atom displacement along of the BiOCl OV (001) direction, lower panel. Insets show the geometries of the systems at the relevant points in the paths.

Supplementary Fig.7 The calculated difference charge density contour of H-BiOCl OV.

Figure 2. Structural characterization of the synthesized powders. (b) Comparison of ¹H NMR spectra of H-BiOCl OV and BiOCl OV; (c) Experimental Fourier transform of the Bi L-edge EXAFS; (c) Raman spectra of all the samples.

Therefore, through theoretical and experimental results, we can be sure the H does enter the original OV of the BiOCl OV photocatalysts and have bonded with the surrounding atoms.

Additionally, we have moved the legend of Figure 2 and 4 under the Figure.

1. Liu, F. et.al. Transfer Channel of Photoinduced Holes on a TiO₂ Surface As Revealed by Solid-State Nuclear Magnetic Resonance and Electron Spin Resonance Spectroscopy. *J. Am. Chem. Soc.* **139**, 10020–10028 (2017).
2. Van de Walle, C. G, and Neugebauer, J. Hydrogen in semiconductors, *Annu. Rev. Mater. Res.* **36**, 179 (2006).
3. Filippone, F, Mattioli, G., Alippi, P., and Amore Bonapasta, A.. Properties of hydrogen and hydrogen–vacancy complexes in the rutile phase of titanium dioxide, *Phys. Rev. B* **80**, 245203 (2009)
4. Janotti, A. and Van de Walle, C. G., Hydrogen multicentre bonds. *Nature Mater.* **6**, 44 (2007) .
5. Koch, S. G., Lavrov, E. V., and Weber. J. Photoconductive Detection of Tetrahedrally Coordinated Hydrogen in ZnO. *Phys. Rev. Lett.* **108**, 165501 (2012).

Reviewer 2

As we all know, the exploration of photocatalytic reaction mechanism associated with defect engineering is a basic issue for designing and developing of photocatalytic semiconductor materials. This paper utilizes a variety of methods to systematically study the mechanism of hydrogenation and synergetic effects among hydrogen atoms and local electronic structures in the defective interlayer BiOCl. This study is profound and systemic, and I will recommend the manuscript for publication with minor revision.

1. Comment

The excellent recyclability and stability of a catalyst should be very important. Please the authors perform relevant tests to verify their as-prepared samples.

Response

According to the referee's suggestion, we added the cycling test of photocatalytic removal of NO in the presence of H-BiOCl OV under illumination by visible light ($\lambda > 420$ nm). As shown in Supplementary Figure 13, H-BiOCl OV can be efficiently reused for NO removal with good recyclability and stability, which did not show any loss of photocatalytic activity. These results clearly show that H-BiOCl OV exhibit good photocatalytic recyclability and stability under visible light.

Supplementary Figure 13 Cycling measurements of photocatalytic NO oxidation with H-BiOCl OV.

Response letter

2. Comment

Please confirm whether the sample structure changes after a long-time test.

Response

We thank for this suggestion, As shown in **Supplementary Fig. 14**, After H-BiOCl OV efficiently reused 5 times for NO removal, the XRD patterns of reused H-BiOCl OV do not show any obvious variation after cycling photooxidation test. These results clearly show that H-BiOCl OV exhibit good photocatalytic stability under visible light.

Supplementary Figure 14 XRD patterns of H-BiOCl OV before and after the cycling test of NO oxidation under visible light.

Response letter

3. Comment

About the EXAFS. The absorption edge of Bi_{4f} maybe also change when the electrons around Bi_{4f} change. Can you supply the raw data before the Fourier transform?

Response

We thank for this suggestion, we have added it as **Fig S4**. Generally, absorption edges shift to higher energies with increasing valence state. There shows a clear shift toward the lower energy side, indicating that a lower state of Bi appears. This is same as the XPS results.

Supplementary Figure 4 XANES spectra at the Bi L_3 -edge.

Response letter

4. Comment

The reaction mechanism is important for catalyst. Can you provide the possible reaction mechanism of NO photocatalytic oxidation for H-BiOCl OV?

Response

We thank for this suggestion. Through the analysis data, the possible reaction mechanism of NO photocatalytic oxidation for H-BiOCl OV can be proposed as follows:

Reaction pathway 1:

Reaction pathway 2:

5. Comment

The authors should compare and review with the relevant data of previously reported literatures of BiOCl OV (Comp. Mater. Sci., 2012, 61: 180-184; ACS Catalysis, 6(12): 8276-8285; J. Mater. Chem. A, 2017, 5(47): 24995-25004; Appl. Catal. B-Environ.: 2018, 228: 87-96, etc.).

Response

We thank for this suggestion; we have added it as reference.

22. Li, H. et al. Oxygen vacancy structure associated photocatalytic water oxidation of BiOCl. *ACS Catal.* **6**, 8276-8285 (2016).

23. Ma, Zh. et al. Oxygen vacancies induced exciton dissociation of flexible BiOCl nanosheets for effective photocatalytic CO₂ conversion. *J. Mater. Chem. A*, **5**, 24995-25004 (2017).

24. Mao, C. et al. Visible light driven selective oxidation of amines to imines with BiOCl: Does oxygen vacancy concentration matter? *Appl. Catal. B* **228**, 87-96 (2018).

28. Zhang, X. et al. Effects of oxygen vacancy on the electronic structure and absorption spectra of bismuth oxychloride. *Comp. Mater. Sci.* **61**, 180-184, (2012).

REVIEWERS' COMMENTS:

Reviewer #1 (Remarks to the Author):

I am satisfied with the response and the modifications made by the authors, and recommend to accept the manuscript for publication as is

Reviewer #2 (Remarks to the Author):

Accept